# Mate selection: A useful approach to maximize genetic gain and control inbreeding in genomic and conventional oil palm (*Elaeis guineensis* Jacq.) hybrid breeding

Billy Tchounke[1], Leopoldo Sanchez[2], Joseph Martin Bell[1], David Cros[3,4]*

**1** Department of Plant Biology, Faculty of Science, University of Yaoundé I, Yaoundé, Cameroon, **2** BioForA, INRAE, ONF, Orleans, France, **3** CIRAD, UMR AGAP Institut, Montpellier, France, **4** UMR AGAP Institut, Univ. Montpellier, CIRAD, INRAE, Institut Agro, Montpellier, France

* david.cros@cirad.fr

**Data Availability Statement:** The simulated datasets and the R script implementing mate selection used during the current study are

## Abstract

Genomic selection (GS) is an effective method for the genetic improvement of complex traits in plants and animals. Optimization approaches could be used in conjunction with GS to further increase its efficiency and to limit inbreeding, which can increase faster with GS. Mate selection (MS) typically uses a metaheuristic optimization algorithm, simulated annealing, to optimize the selection of individuals and their matings. However, in species with long breeding cycles, this cannot be studied empirically. Here, we investigated this aspect with forward genetic simulations on a high-performance computing cluster and massively parallel computing, considering the oil palm hybrid breeding example. We compared MS and simple methods of inbreeding management (limitation of the number of individuals selected per family, prohibition of self-fertilization and combination of these two methods), in terms of parental inbreeding and genetic progress over four generations of genomic selection and phenotypic selection. The results showed that, compared to the conventional method without optimization, MS could lead to significant decreases in inbreeding and increases in annual genetic progress, with the magnitude of the effect depending on MS parameters and breeding scenarios. The optimal solution retained by MS differed by five breeding characteristics from the conventional solution: selected individuals covering a broader range of genetic values, fewer individuals selected per full-sib family, decreased percentage of selfings, selfings preferentially made on the best individuals and unbalanced number of crosses among selected individuals, with the better an individual, the higher the number of times he is mated. Stronger slowing-down in inbreeding could be achieved with other methods but they were associated with a decreased genetic progress. We recommend that breeders use MS, with preliminary analyses to identify the proper parameters to reach the goals of the breeding program in terms of inbreeding and genetic gain.

available from PalmElit SAS (palmelit@palmelit.com, PalmElit SAS, Bat 14, Parc Agropolis – 2214 Bd de la Lironde, 34980 Montferrier sur Lez – France, https://www.palmelit.com/).

**Funding:** This work was partly funded by a grant from PalmElit SAS (Méthodes de SAM) which supported BT and a grant from INRAE SELGEN (Breed2Last) which supported DC. The funders had no role in study design, data collection and analysis, decision to publish, or preparation of the manuscript.

**Competing interests:** The authors have declared that no competing interests exist.

## Author summary

Oil palm is a tropical perennial plant and the world main source of vegetable fats. Oil palm breeding requires the estimation of the additive genetic value of the selection candidates to identify the best hybrid crosses, that will be released as commercial varieties. Genomic selection (GS) has the potential of boosting the rate of genetic progress for quantitative traits. However, GS could also be used in conjunction with approaches to optimize selection and mating, in order to maximize genetic progress while limiting inbreeding, which can increase faster with GS and can have negative consequences (inbreeding depression detrimental for seed production, loss of favorable alleles). This study compared several strategies to reach this goal. In particular, we considered mate selection (MS), a method that uses an algorithm optimizing the selection of individuals and their crosses. Our results showed that MS reduced inbreeding in parental populations and increased annual genetic progress. The originality of this work resides in the fact that, despite the high computational burden of MS, we applied it in a large simulated dataset and showed its efficiency in the context of genomic and conventional selection. Also, this is the first study to characterize, in terms of breeding rules, the optimized solution (i.e. the set of selected individuals and their mating design) retained by MS.

## Introduction

Genomic selection (GS) is an effective method for improving quantitative traits [1]. It is based on dense marker coverage of the genome and on statistical methods capable of simultaneously valuing the information of all markers to estimate the genetic value of candidates for selection. The GS model is calibrated on a set of evaluated and genotyped individuals (training population) and is applied to selection candidates that have not been evaluated but genotyped with the same markers. GS is therefore particularly interesting when conventional, i.e. phenotypic selection (PS) is limited by the evaluation phase. This makes GS of special interest for perennial crops, with typically long breeding cycles (>10 years) and a limited capacity to assess numerous candidates. For perennials, several studies have already shown the benefits of GS, for example, in eucalyptus and conifers [2], apple [3], coffee [4] and rubber tree [5]. However, GS can also increase the rate of inbreeding compared to PS, notably per time unit. This was reported both in animals [6] and plants [7]. Inbreeding corresponds to the mating of related individuals [8,9]. This is unavoidable in populations of finite sizes and can be exacerbated by selection and high selection accuracy. Inbreeding causes allelic fixation, which reduces additive variance and therefore the rate of genetic progress, and can generate inbreeding depression [8,9]. Inbreeding depression corresponds to a reduction in the mean value of a trait, and it can be observed for traits where dominance effects are involved, as a result of the reduction in the proportion of heterozygote loci. Indeed, the latter causes the expression of detrimental recessive mutations in homozygous genes (partial dominance hypothesis). It also reduces the proportion of heterozygous genes with over-dominance effects, as well as the probability of having epistatic effects between dominant effects at heterozygous genes. Besides, a slow increase in inbreeding is more efficient to select against deleterious recessive mutations (purging), than a fast increase in inbreeding [7]. Methods of inbreeding management are therefore required, in particular in the context of GS.

Various strategies have been developed to limit the increase in inbreeding over generations while optimizing the rate of genetic gain, in particular mate selection (MS) [10–13], which uses an optimization algorithm to do simultaneously the selection of candidates and the

allocation of mates among the selected ones. Unlike truncation selection, where the best individuals, selected as their genetic value is beyond a certain value, are randomly mated, the MS approach also has the potential to increase the genetic gain by identifying optimal matings between individuals that are complementary in terms of genetic characteristics. However, few studies were carried out so far on MS, in particular in plant species [14]. This is likely due to the complexity of the practical implementation of controlled crosses in many species and, to a lesser extent, to the high computational burden of the method. Also, to our knowledge, there is so far no study translating, in terms of breeding decisions (i.e. number of matings per selected individuals, proportion of selfings, etc.), the rules implicitly followed by the algorithm to converge towards an optimized solution.

In the present study, we focused on one of those perennial crops where GS appeared to have a great potential, the world major oil crop, the oil palm (*Elaeis guineensis* Jacq.) [15]. Conventional oil palm breeding relies on recurrent reciprocal selection (RRS), where two complementary populations are crossed to generate commercial varieties with hybrid vigor (heterosis) on the production of bunches. Production of bunches is a multiplicative trait where heterosis can be explained by a model without dominance, by the product of purely additive and complementary components, bunch number and bunch weight. Studies showed that GS would increase the performances of oil palm hybrids and of clonal varieties selected within hybrids by increasing selection intensity and shortening breeding cycles [15,16]. However, it also showed that GS would increase the rate of inbreeding in the parental populations [17]. This would reduce the potential of long-term genetic progress and could generate inbreeding depression in the parents. Indeed, inbreeding depression was reported in oil palm [18,19], causing poor germination, abnormal seedlings, abortive bunch formation, poor fruit set and reduced yield, height and leaf area. Inbreeding depression in parental populations is not a problem in terms of hybrid performance (as hybrid crossings restore a high level of heterozygosity), but it might be detrimental for reproductive traits that affect seed production (e.g. seed germination). So far, to our knowledge, there is no study published on inbreeding management and on the optimization of matings among selected individuals in a case such as oil palm, where the aim is to maximize genetic gain in hybrids on a multiplicative trait while controlling inbreeding in the parental populations.

The goal of the present study was therefore to understand the benefits and downsides of MS and several methods of inbreeding management in terms of the performance of oil palm cultivars and the inbreeding in the parental populations, over several generations of GS and PS, and in order to help oil palm breeders to choose the most relevant breeding scheme. In addition, this study aimed at deciphering the eventual emerging rules revealed by the use of the optimization algorithm, in terms of selection and mating. Practical reasons make that this cannot be studied empirically, as there are many scenarios to compare, and because the perennial nature of oil palm makes that a considerable number of years would be necessary for each scenario, with a generation of oil palm breeding requiring up to 20 years. In such a case, computer simulations become the only possible approach. Simulations are particularly useful for breeding method comparison [20,21].

Simulation algorithms belong to two categories, backward-in-time and forward-in-time approaches, with characteristics that make them suitable for different problems [20,21]. The backward-in-time approach, or coalescent approach, starts from the observed sample in the present generation and works backward to trace all alleles to a single common ancestor, and then works forwards to the current generation, adding mutations or other genetic information into the simulated genealogy. This approach is particularly adapted for studies on an evolutionary timescale and with minor deviations from the Wright-Fisher model. The forward-in-time approach is centered on individuals. This makes it slower and leads to the requirement of

initial conditions of genetic variation, but this also gives it the ability to model complex scenarios. As a consequence, forward-in-time simulations are particularly suitable to study the effect of a limited number of generations of artificial selection in well-characterized species for which the available data allow defining the initial genetic conditions.

Here, we opted for a forward-in-time simulation approach, where the evolution of oil palm populations was simulated over four generations, through the simulation of haplotypes and meiosis, and implementing MS and various inbreeding management methods with GS and PS. As obtaining the results in a reasonable amount of time remained challenging, in particular as we considered MS, we used a high-performance computing cluster and parallel computing. The genetic progress for total bunch weight and inbreeding in the parental populations were compared between breeding scenarios that relied on: (i) the conventional method (i.e. with truncation selection, random mating of selected individuals and no inbreeding management), (ii) MS and (iii) five simple methods of inbreeding management (i.e. limiting deterministically the number of full-sibs selected, prohibiting selfings and combining the two last approaches).

## Materials and methods

### Simulation overview

The target character of the selection is the total bunch weight (FFB, for fresh fruit bunch, expressed in kg per palm). It is a multiplicative character equal to the product of the bunch number (BN) and the average bunch weight (BW), two mostly additive traits with a strong negative correlation [18,22]. Heterosis in FFB is a case of heterosis for multiplicative traits [23,24]. As BN and BW have mostly additive inheritance, and as inbreeding depression is of concern for other characters involved in seed production (e.g. seed germination, abortive bunch formation or poor fruit set in the Deli population, used as mother palms), this simulation study relies on the simulation of BN and BW following additive genetic architecture, with the FFB values being deduced from BN and BW. Other authors also studied the management of inbreeding by additive simulations, in plants and animals (e.g. [7,25,26]).

Different breeding scenarios were considered. They consisted in combinations of types of breeding schemes (reciprocal recurrent phenotypic selection, RRS, and reciprocal recurrent genomic selection, RRGS) and of methods to deal with selection and mating among selected individuals: conventional method, MS and simple methods of inbreeding management. For computational reasons, MS and simple methods of inbreeding management were only implemented in the La Mé population, Deli population being submitted to the conventional method.

Custom scripts written with the R software version 3.6.3 was used for simulations and analyses [27]. The ASReml-R package was used to implement the mixed model analyses by the best linear unbiased prediction methodology (BLUP) [28,29]. The HaploSimR package was used to implement a forward simulation approach and to simulate haplotypes and meiosis [30]. It takes advantage of the sparse nature of genotypic data to optimize memory usage. The R packages snow [31] and doSNOW [32] were used to implement parallel computing.

### Initial populations

The initial Deli and La Mé parental populations constituting the starting point of the study (generation 0) were generated following the simulation procedure described in [33], with calibration using real values taken from the actual Deli and La Mé breeding populations used by PalmElit and its partners, and from the literature (Table 1).The simulated initial La Mé population had 24 founders representing the bottleneck event at the origin of the actual population in Côte d'Ivoire in the 1920s. This was followed by two generations with an increasing number of

**Table 1. Genetic parameters in the initial Deli, La Mé and hybrids populations obtained by simulation.** Values are means of 30 replicates ± SD. Real values used as targets to calibrate the simulations: Fst = 0.55, $h^2$ = 0.56, r(BN, BW) = -0.9 (Deli); -1.0 (La Mé), interpopulation additive variances [52] = 2.66 (BN), 0.30 (BW) in Deli; 1.92 (BN), 0.22 (BW) in La Mé, molecular inbreeding = 0.79 in Deli; 0.73 in La Mé. For LD, the calibration was made on the profile of the LD curves.

| | | Population | | |
|---|---|---|---|---|
| Parameter | Trait | Deli | La Mé | Deli x La Mé |
| *Fst* | | 0.54 ± 0.01 | | |
| *LD (cM)* [1] | | 8.42 ± 0.3 | 4.45 ± 0.2 | |
| *FFB* | | 188.88 ± 14.15 | 197.79 ± 15.7 | 221.35 ± 15.67 |
| *heterosis (in % parental FFB)* | | | | 14.52 ± 4.62 |
| *$h^2$* | BN | 0.54 ± 0.04 | 0.48 ± 0.05 | |
| | BW | 0.55 ± 0.03 | 0.47 ± 0.05 | |
| *True breeding values* | BN | 9.22 ± 0.77 | 19.73 ± 1.88 | |
| | BW | 20.52 ± 0.85 | 10.07 ± 0.83 | |
| *Genetic correlation r(BN, BW)* | | -0.76 ± 0.04 | -0.75 ± 0.05 | |
| *Interpopulation additive variance* | BN | 2.74 ± 0.54 | 2.15 ± 0.49 | |
| | BW | 0.36 ± 0.06 | 0.27 ± 0.05 | |
| *Mean interpopulation additive variance at QTL (in % total)* | BN | 0.25 ± 0.04 | 0.27 ± 0.06 | |
| | BW | 0.24 ± 0.04 | 0.28 ± 0.04 | |
| *Genealogical inbreeding* | | 0.31 ± 0.03 | 0.13 ± 0.02 | |
| *Molecular inbreeding* | | 0.72 ± 0.01 | 0.73 ± 0.01 | |
| *Annual genetic progress in cycles -2 and -1 (in % hybrid FFB at generation -2)* | | | | 0.49 ± 0.08 |

[1] mean distance (cM) where linkage disequilibrium (LD) measured by $r^2$ between adjacent loci was 0.1

individuals (70 and 100 individuals per generation), with mass selection for FFB (see [33] for more details). Similarly, the simulated initial Deli population had four founders representing the bottleneck event corresponding to the original four oil palms planted in Indonesia in 1848. This was followed by eight generations with an increasing number of individuals (25, 75, 75, 75, 75, 75, 75 and 100), with mass selection for FFB from the second generation and selfings allowed in the last two generations.

Thirty sets of Deli and La Mé initial populations were generated, to be used as replicates for the simulation. 800 QTLs were simulated per trait, including 70% of QTLs with pleiotropic effects on BN and BW, assuming pleiotropy played a role in the negative correlation between the two traits. QTLs effects were sampled from normal distribution and unknown positions of QTLs were sampled randomly from segregating SNP with a minor allele frequency (MAF) above 0.1. The resulting genetic correlation between the traits BN and BW was -0.76 ± 0.04 for Deli and -0.75 ± 0.05 for La Mé. The mutation rate for QTLs and single nucleotide polymorphisms (SNPs) markers was set to zero. The phenotypes for BN and BW were simulated as the sum of mean value of the population, additive breeding value and environmental effects. The environmental effects on BN and BW were simulated from normal distributions with mean zero and residual variances $\sigma^2_{e(BN)}$ and $\sigma^2_{e(BW)}$. These residuals variances for each trait were obtained from heritability $h^2$ by the formulas $\sigma^2_{e(BN)} = \sigma^2_{a(BN)}(1 - h^2)/h^2$ and $\sigma^2_{e(BW)} = \sigma^2_{a(BW)}(1 - h^2)/h^2$, were $\sigma^2_{a(BN)}$ and $\sigma^2_{a(BN)}$ are genetic variances on BN and BW respectively. All values of the genetic parameters in the initial Deli, La Mé and hybrid populations obtained by simulation are given in Table 1.

### Breeding schemes

From generation 0, four cycles of reciprocal recurrent selection were simulated in its conventional phenotypic version (RRS) and in a genomic version (RRGS) to increase FFB in Deli x La Mé hybrids. The RRS and RRGS breeding schemes are explained in more detail in [17,33] and shown here in Fig 1. RRS requires progeny tests at each generation while in RRGS there were progeny tests every two generations (first and third generations). For RRS, the individuals are selected within the two populations based on their estimated breeding values in hybrid crosses with the other population (EBVs) obtained from the hybrid progeny tests (see next section). For RRGS, the individuals are selected similarly, except that GEBVs (genomic EBVs, i.e. obtained with the genomic model) are available for the progeny-tested and the non-progeny-tested individuals.

With RRS, the progeny tests include 120 parents per population and 20 hybrid individuals per cross, with each parent being crossed randomly with, on average, 2.1 parents of the other population (i.e. incomplete factorial mating design). At each generation, 16 individuals are selected per population and are mated within populations to produce the new generation of selection candidates (see next section for the different methods compared here), comprising 120 individuals per population. The EBVs of the progeny tested parents are obtained by linear mixed model analysis with pedigree-based BLUP[29], the pedigree of the starting individuals (generation 0) being known over four previous generations for the Deli and two generations for the La Mé. A cycle of RRS lasts 19 years.

With RRGS, the progeny tests are made in the same way as for RRS. The GEBVs are obtained with GBLUP, using 2,250 SNP markers to compute the matrix of genomic coancestries. They were neutral SNPs (i.e. not in QTLs) and randomly sampled among the existing SNPs. The model is trained with the data of the most recent progeny test. The number of selected individuals per population and cycle is 16, as in RRS. They are mated with the same methods as in RRS to generate the next generation. Two RRGS scenarios were simulated, considering 120 and 330 selection candidates per generation and population. A GS breeding cycle without progeny tests lasts 6 years.

The whole process is repeated 30 times, using each time one of the 30 replicates of initial populations.

## Methods of selection and mating

### Conventional method

In each population, the top 16 individuals, i.e. giving hybrid crosses with the highest expected bunch production, were selected. The expected bunch production of the hybrid crosses was calculated as the product of the mean parental EBVs/GEBVs for BW and BN. The selected individuals are then crossed four times with partners chosen randomly, self-fertilization being authorized and without reciprocal crosses. This gave an incomplete diallel mating design with 32 crosses.

### Mate selection

Mate selection (MS) jointly optimizes selection and mating within populations. The optimization aims at maximizing genetic progress while imposing a limit (defined by the user) on the increase in inbreeding. MS is used here to retain an $X$ matrix of matings among the La Mé selection candidates in order to obtain the progenies with the highest expected hybrid FFB in crosses with Deli and an inbreeding level below the fixed threshold. The $X$ matrix comprises elements 0s (crosses not made) and 1s (crosses made),

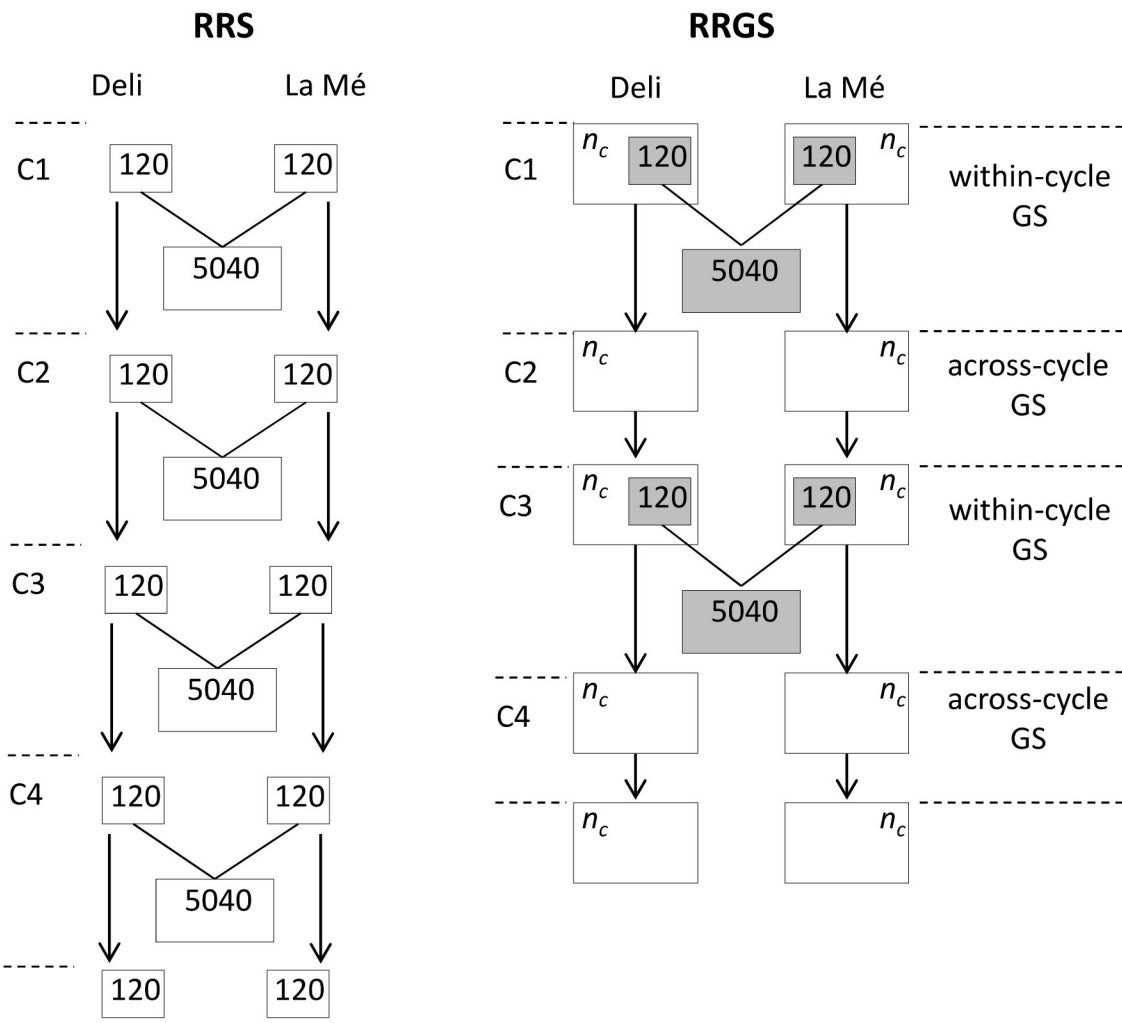

**Fig 1. Reciprocal recurrent phenotypic selection (RRS) and reciprocal recurrent genomic selection (RRGS) between Deli and La Mé oil palm breeding populations.** C1. C2, C3, C4: breeding cycles (C1 starting with parental individuals of generation 0). $n_C$: number of selection candidates for RRGS (120 or 330). Grey boxes: training set for GS model. Figures in boxes indicate numbers of individuals.

MS can be implemented using the simulated annealing (SA) optimization algorithm [11]. SA mimics the slow cooling that leads to a perfect crystalline state, i.e. the minimum energy state, in a metal annealing process [34,35]. Starting from an initial random solution at high temperature, several alternative solutions derived from the initial solution (neighbor solutions, generated by a replacement function) are evaluated over the algorithm iterations, until a new solution is accepted. The evaluation of the alternative solutions is made by an evaluation function $E$ and the decision of the acceptance or rejection of the alternative solution is based on the difference between the evaluation values of the initial solution and of the alternative solution ($DELTA_E = E_{alternative} - E_{initial}$). If $DELTA_E < 0$ the alternative solution is accepted. If $DELTA_E \geq 0$ the algorithm can also accept the alternative solution with a probability $P = e^{\frac{-DELTA_E}{T}}$, to avoid being trapped in a local minimum. Once a new solution is accepted, the temperature is reduced, and as the temperature decreases, the probability of accepting an alternative solution that does not decrease the evaluation value tends to zero, so the algorithm converges to an optimal solution.

At the start of the SA algorithm, the 60 best La Mé individuals are preselected based on their EBVs/GEBVs, as described in the conventional method. The initial solution is generated by randomly selecting 16 individuals from the 60 preselected La Mé and by randomly mating them (with selfing allowed and reciprocals not allowed) in 32 crosses. This initial solution is materialized by the upper triangular part and diagonal of matrix $X$, with the 60 La Mé individuals in rows and columns and values 1s and 0s indicating crosses made and not made, respectively. For Deli, a 16 $x$ 16 matrix is also generated, indicating the mating design among the selected individuals, obtained according to the conventional breeding method. The initial matrix generated for the Deli is used throughout the SA process.

The evaluation function $E$ is definedas the opposite of the mean expected FFB of the hybrids obtained by crossing the progenies of the selected Deli and La Mé individuals. Let $D_iD_j$ be the progenies from the cross between Deli $i$ and $j$, $LM_iLM_j$ from the cross between La Me $i$ and $j$, and $FFB_{D_iD_j \times LM_iLM_j}$ the expected FFB performance of the $D_iD_j \times LM_iLM_j$ hybrid cross. The evaluation function is:

$$
\begin{aligned}
E = -&\left[ 0.5\left( g_{D_{i(BN)}} + g_{D_{j(BN)}} \right) + 0.5\left( g_{LM_{i(BN)}} + g_{LM_{j(BN)}} \right) \right] \\
&\times \left[ 0.5\left( g_{D_{i(BW)}} + g_{D_{j(BW)}} \right) + 0.5\left( g_{LM_{i(BW)}} + g_{LM_{j(BW)}} \right) \right],
\end{aligned}
$$

were $g$ corresponds to the EBV/GEBV.

The replacement function proceeds as follows: from a given $X$ solution matrix, it removes $nr$ crosses made and makes $nr$ new crosses (i.e. switching randomly $nr$ 1 and $nr$ 0 in $X$). Each time the replacement function is called, the switch is repeated until a valid alternative solution is found, i.e. respecting the rules constraining $\{X\}$ (see below), or until the number of alternative solutions tested reached 5,000. In the latter case, the input $X$ is kept as the final solution and the SA stops (no convergence but no better solution found). The value of $nr$ was 3 in the first five iterations, then 2 until iteration 15 and finally 1. This way, the number of crosses that are randomly replaced decreases progressively as the space of possible$X$ narrows. The rules constraining $\{X\}$ at generation $n$, designated here by $X = (x_{ij})$, with $j \geq i$, were:

$$
i) \ x_{ij} = 0 \text{ or } x_{ij} = 1
$$

$$
\sum_i \sum_j x_{ij} = 32
$$

$$
\frac{1}{32} \sum_i \sum_j x_{ij} f_{ij} < F(n) + \Delta F(n)
$$

and $f_{ij}$ denotes the coefficient of kinship between individuals $i$ and $j$, $F(n) + \Delta F(n)$ is the inbreeding threshold in the progenies of the selected La Mé and $F(n)$ is the average inbreeding in the 120 La Mé selection candidates. To improve the approach described in Toro and Perez-Enciso [10]and Sanchez et al.[11], the inbreeding threshold was defined directly by the script at each generation, so that the level is not too high, to ensure inbreeding is constrained compared to the conventional method, nor to low, to ensure valid solutions exist. The formula for the threshold is given by:

$$
\Delta F(n) = (Frand(n+1) - F(n)) + c_{\Delta F}(Fconv(n+1) - Frand(n+1)).
$$

$F(n)$ is, in RRS, the mean genealogical inbreeding of the La Mé individuals of generation $n$ and, in RRGS, their mean genomic inbreeding, i.e. the mean diagonal of the $G$ matrix of the La

Mé individuals of generation $n$ (computed according to [36]) minus 1. $Frand(n + 1)$ is the mean inbreeding over 12 replicates of La Mé progenies obtained by the random mating of randomly selected La Mé individuals of generation $n$, and is considered as the lower bound in the increase in inbreeding in generation $n + 1$ for MS. In RRS, $Frand(n + 1)$ uses genealogical inbreeding, while in RRGS it uses the genomic inbreeding, computed as the genomic coancestry according to [36] between the mated parents. $Fconv(n + 1)$ is the mean inbreeding over seven replicates of La Mé progenies obtained by the conventional method of selection and mating among the La Mé individuals of generation $n$, and is considered as the upper bound in the increase in inbreeding for MS. In RRS, $Fconv(n + 1)$ uses genealogical inbreeding and, in GS, genomic inbreeding. The use of genealogical information in RRS against genomic information in GS came from the fact that, to control inbreeding, it must be measured on the same basis as what is used to estimate breeding values [25]. Finally, the user can introduce different levels of stringency in terms of inbreeding management by the coefficient $c_{\Delta F}$. Three values of $c_{\Delta F}$ were considered: 0%, 25% and 50%, corresponding to decreasing levels of stringency in terms of inbreeding management.

The lowering of the temperature along iterations $i$ is made according to the law of decay:

$$T_{i+1} = 0.95 \times T_i.$$

The initial temperature $T_0$ was determined so that the probability $P$ of accepting an alternative solution to the initial solution $X_0$ that did not reduce the evaluation value of $X_0$ ($E_0$) was 0.5. This was computed on average over 10 replicates, i.e. 10 alternative solutions to $X_0$. In case no valid alternative solutions to $X_0$ could be found over the 10 replicates, a new $X_0$ was generated, with the process having up to five successive $X_0$ in case of failure. In case $T_0$ was $\leq 0$, $T_0$ was set to 0.15.

Table 2 gives the values used for the different SA parameters according to the number of La Mé selection candidates. These values were defined by trial and error in preliminary analyses, considering computation time and convergence (showing e.g. that using higher $n_{presel}$ was useless, as it increased computation time while the method did not select individuals ranked after the $n_{presel}$ values used). S1 Fig gives a visual example of an optimization path according to iterations and temperature levels. $T$ decreases when the number of iterations $n_2$ is reached or when the maximum number of solutions accepted at a given temperature $n_M$ have been accepted. The convergence was reached when no solution was accepted at a given $T$.

The SA algorithm is launched on a high-performance computing data center in parallel on 96 cores, starting with random initial solutions $X_0$. Finally, the optimal solution retained was the solution with the lowest evaluation value over the 96 cores.

**Table 2. Parameters for the simulated annealing algorithm.** $n_{presel}$ number of preselected La Mé selection candidates, $n_1$ number of iterations to reduce temperature ($T$), $n_2$ number of iterations at a given $T$ and $n_M$ maximum number of solutions accepted at a given $T$.

| | Number of La Mé selection candidates | |
|---|---|---|
| | 120 | 330 |
| $n_{presel}$ | 50 | 60 |
| $n_1$ | 80 | 100 |
| $n_2$ | 50 | 100 |
| $n_M$ | 30 | 60 |

### Simple methods of inbreeding management

The conventional method was extended to implement five simple methods of inbreeding management: prohibiting self-fertilization (NoSelf), by imposing a threshold on the number of individuals selected per family (i.e. full-sibs) set to one (FS_T1) and three (FS_T3) (with selfings remaining allowed), and combining the two methods (FS_T1_NoSelf and FS_T3_NoSelf).

### Analysis of results

The inbreeding coefficient is the probability that two alleles in an individual are identical by descent relative to a founder population where all alleles are assumed unrelated [37]. Three measures of inbreeding were made in the La Mé population. The genealogical inbreeding was computed from the pedigree starting from the generation of the 24 founders using the function calcInbreeding of the pedigree R package [38]. The genomic inbreeding was measured on the SNPs that were polymorphic in the initial La Mé population. The SNPs used were all neutrals (i.e. not in QTLs) and were 13,779 on average over the 30 replicates, ranging from 13,144 to 14,244 according to replicates. Inbreeding at SNPs was computed for each La Mé individual as the percentage of homozygote SNPs. The genomic inbreeding was then computed as the mean value over the individuals comprising the population. Inbreeding was measured similarly at QTLs.

The genetic progress was measured in terms of hybrids production (FFB) at the end of cycle 4, and expressed in percentage of FFB at generation 0, i.e. $100 \times (FFB_4 - FFB_0) / FFB_0$, with $FFB_4$ the FFB of the hybrids between the progenies of the Deli and La Mé individuals selected at the end of the fourth cycle and $FFB_0$ the FFB of the hybrids between the Deli and La Mé individuals used as selection candidates in generation 0.

To gain a better understanding of how the optimization procedure used for MS, we calculated, in the La Mé population, the following parameters characterizing the solutions selected by MS (i.e. the set of selected individuals and their mating design): percentage of self-fertilization, rank of selected individuals (with the individuals with the highest expected bunch production in hybrid crosses having the lowest rank value), number of crosses per selected individual, number of individuals selected per full-sib family, correlation between rank and number of crosses of the selected individuals, correlation between the ranks of the individuals crossed, mean rank of the self-fertilized selected individuals and mean relationship between the La Mé individuals crossed with each other to produce the next generation.

## Results

### Inbreeding measured from pedigree and SNPs

Inbreeding measured at SNPs and genealogical inbreeding gave similar trends along generations, across methods of selection and mating and across breeding schemes (see Fig 2 for mean values of inbreeding computed from SNPs, S2 Fig for variations of inbreeding computed from SNPs among replicates and S3 Fig for mean values of genealogical inbreeding). However, the inbreeding values at SNPs were much higher than the genealogical inbreeding, with mean SNP-based inbreeding ranging from 0.73 to 0.87 and mean genealogical inbreeding ranging from 0.10 to 0.47, depending on generation, breeding scheme and methods of selection and mating.

Without inbreeding management (conventional method), there was a strong increase in inbreeding in the La Mé population along the generations of phenotypic and genomic selection. When using MS or the simple methods of inbreeding management, inbreeding also

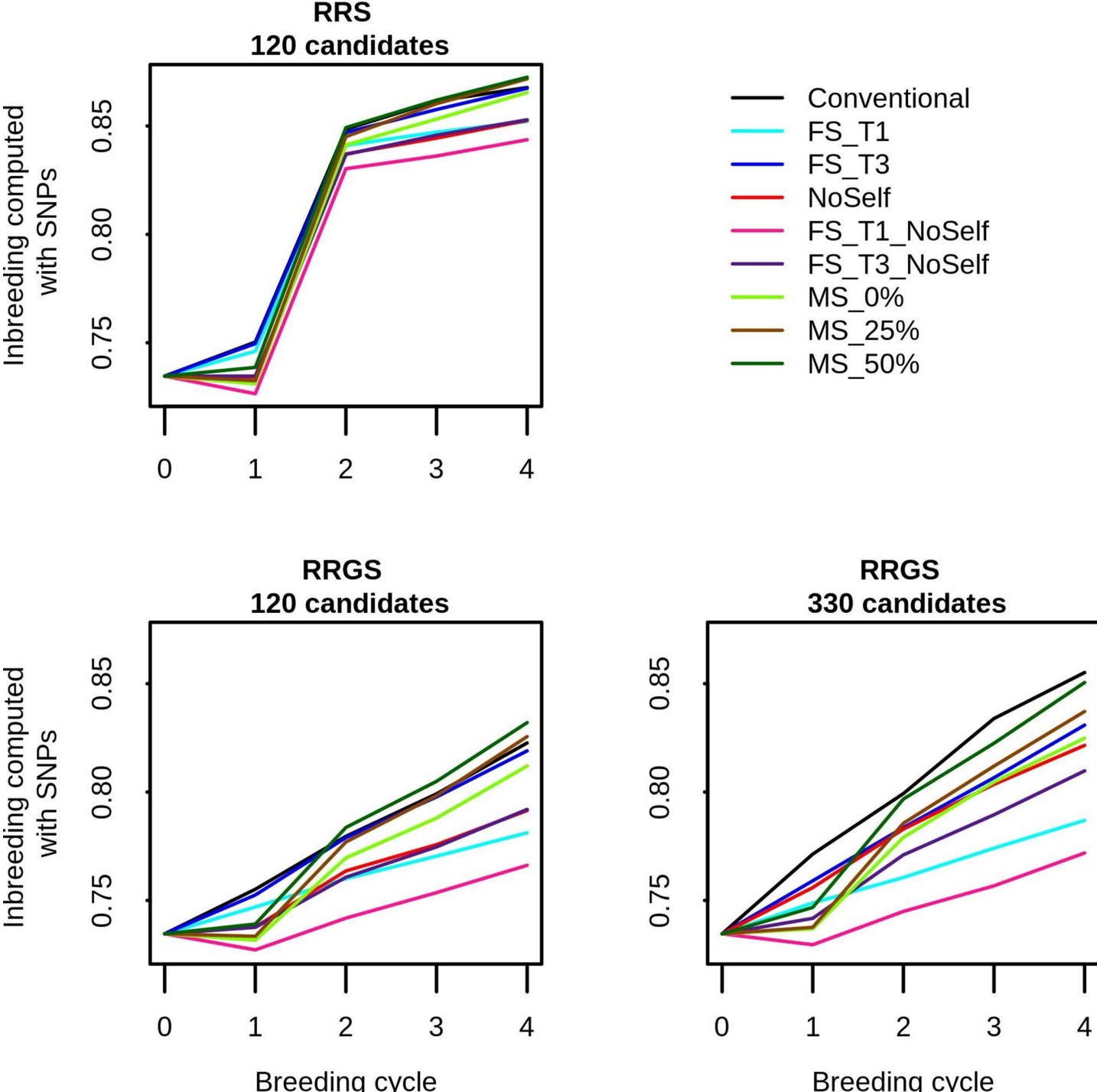

**Fig 2. Inbreeding computed with the SNPs in the La Mé population according to the generations (0–4), breeding schemes (RRS with 120 candidates, RRGS with 120 candidates and RRGS with 330 candidates) and methods of selection and mating.** Figures are means over 30 replicates.

increased over the four generations, but usually less than without inbreeding management. FS_T1_NoSelf was the most efficient method to mitigate the increase in inbreeding. With mate selection, the increase in inbreeding was, as expected, affected by the coefficient $c_{\Delta F}$ used

to control the inbreeding threshold: with MS_0%,inbreeding was always lower than with the conventional method; with MS_25%, inbreeding in the first generation was lower than with the conventional method but the gap filled in the subsequent generations (except in RRGS with 330 candidates where inbreeding remained markedly lower than with the conventional method) ; and with MS_50% a similar trend was observed compared to MS_25% but with higher values of inbreeding (which could even become higher than with the conventional method, in particular with 120 selection candidates). After four breeding cycles, the change in inbreeding computed from the SNPs with MS compared to the conventional method ranged from +0.6% with MS_50% in RRS to -3.6% with MS_0% in RRGS with 330 candidates (Fig 3). When measured with the pedigree, the differences were exacerbated and ranged from +10.3% with MS_50% in RRS to -29.7% with MS_0% in RRGS with 330 candidates (S3 Fig). The differences were significant in some cases, in particular with GS and 330 candidates. All the simple methods of inbreeding management led to significant reductions in the increase in inbreeding after four breeding cycles, except FS_T3 when 120 selection candidates were used. The increase in inbreeding was always significantly higher when the maximum number of individuals selected per full-sib family was three against one. For a given threshold in the number of individuals selected per full-sib family, the increase in inbreeding was always reduced when selfings were not allowed, and this reduction was almost always significant. Simply prohibiting self-fertilization significantly reduces the inbreeding increase compared to the conventional method. With genomic selection, inbreeding increased with the number of selection candidates. This resulted from the fact that the number of crosses was fixed (32), making that a larger population of selection candidates led to larger families of full-sibs, which gave the possibility of selecting more full-sibs and thus to reach higher levels of inbreeding.

SNP-based inbreeding after four generations was higher for RRS compared with RRGS using 120 candidates. However, the difference was moderate (<10%) while the decrease in number of years (76 in RRS against 50 in RRGS) was strong, leading to higher inbreeding per year with RRGS.

## Inbreeding at QTLs

When inbreeding was measured at QTLs, the pattern of increase along generations and across methods of selection and matings and breeding schemes was very similar to when measuring inbreeding from the SNPs (see S5 Fig with the example of QTLs controlling trait 1). However, inbreeding was higher when measured at QTLs than at SNPs, with inbreeding measured at QTLs of trait 1 ranging from 0.88 to 0.95, against values ranging from 0.73 to 0.87 with SNPs, depending on generation, breeding scheme and methods of selection and mating.

## Genetic progress

In all breeding schemes, mate selection always increased the genetic progress compared to the conventional method, with an increase that was usually significant after four cycles (Figs 4 and 5 and S6). It also generally outperformed all the other methods of selection and mating, in particular with 120 candidates, where mate selection gave the highest genetic progress with all values of $c_{\Delta F}$. The genetic progress was on average higher when $c_{\Delta F}$ increased, i.e. when the constraint in terms of inbreeding was relaxed. Compared to conventional selection, mate selection increased the annual genetic progress after four cycles from 3.9% with MS_0% in RRGS with 330 candidates to 13.6% with MS_0% in RRS with 120 candidates. Among the simple methods of inbreeding management, FS_T1 and FS_T1_NoSelf gave the lowest genetic progress, while FS_T3, FS_T3_NoSelf, NoSelf and the conventional method led to intermediate genetic progress.

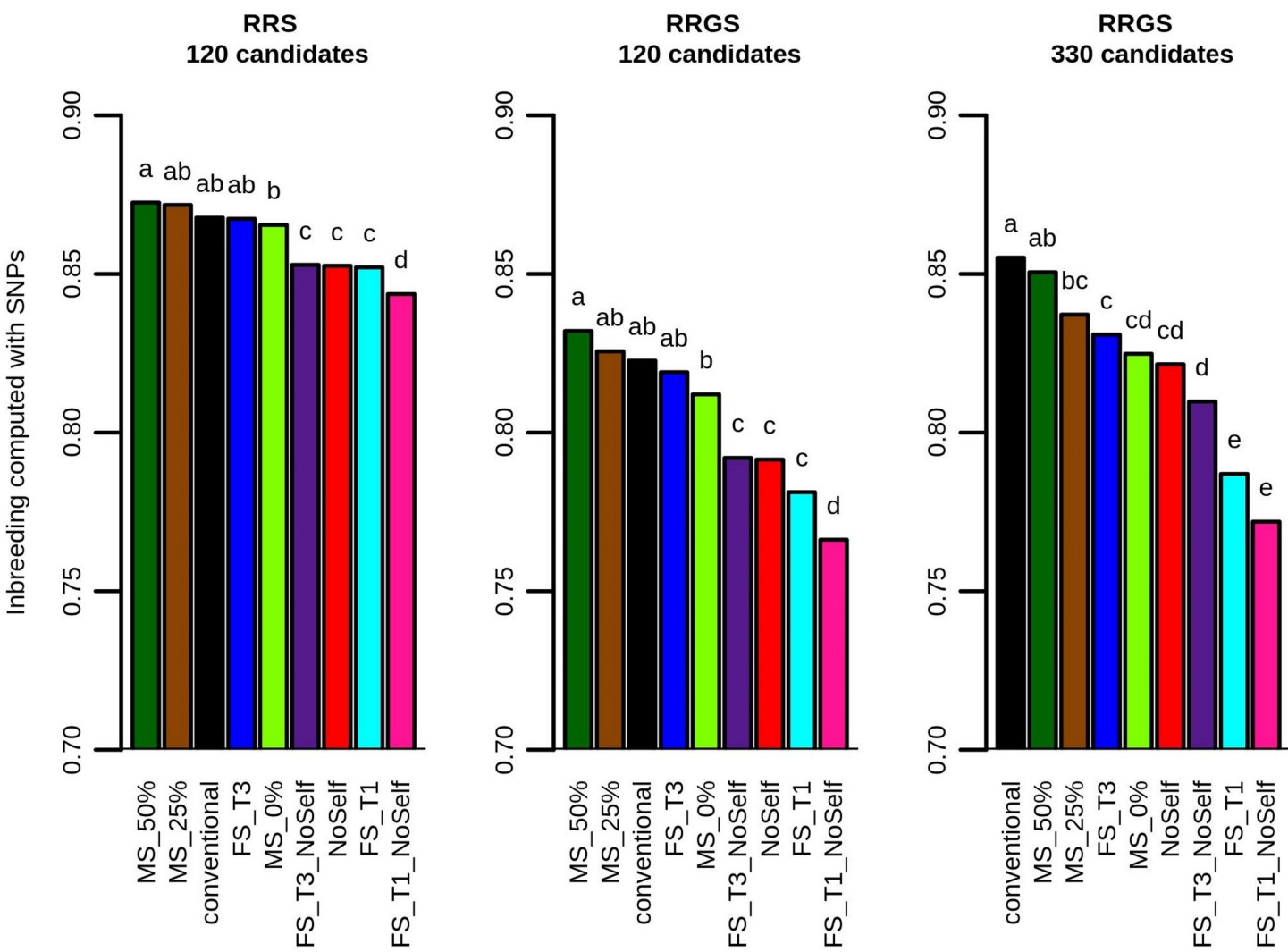

**Fig 3. Inbreeding computed with the SNPs in the La Mé population after four breeding cycles according to method of selection and mating and breeding scheme (RRS with 120 candidates, RRGS with 120 candidates and RRGS with 330 candidates).** Figures are means over 30 replicates. Values with the same letters are not significantly different within a breeding method at P = 5%.

The genetic progress for FFB after four cycles was higher with RRS than with RRGS (Fig 4), but RRGS outperformed RRS in terms of annual genetic progress, in particular with 330 selection candidates (Fig 5).

## Selection and mating

Large variations were found in the percentage of self-fertilization in the selected La Mé individuals according to the method of selection and mating (Fig 6A). With the conventional method, FS_T1 and FS_T3, around 12% of crosses among the selected individuals were selfings, as expected under random mating. With the other methods, the percentage of selfings was lower.

Regarding the maximum rank of selected individuals, all MS scenarios reached values higher that 16, the number of selected parents (Fig 7). The maximum rank of selected individuals was greatest when the constraint on the number of individuals per full-sib family was the strongest (FS_T1, FS_T1_NoSelf). As expected, as using 330 candidates in RRGS led to larger

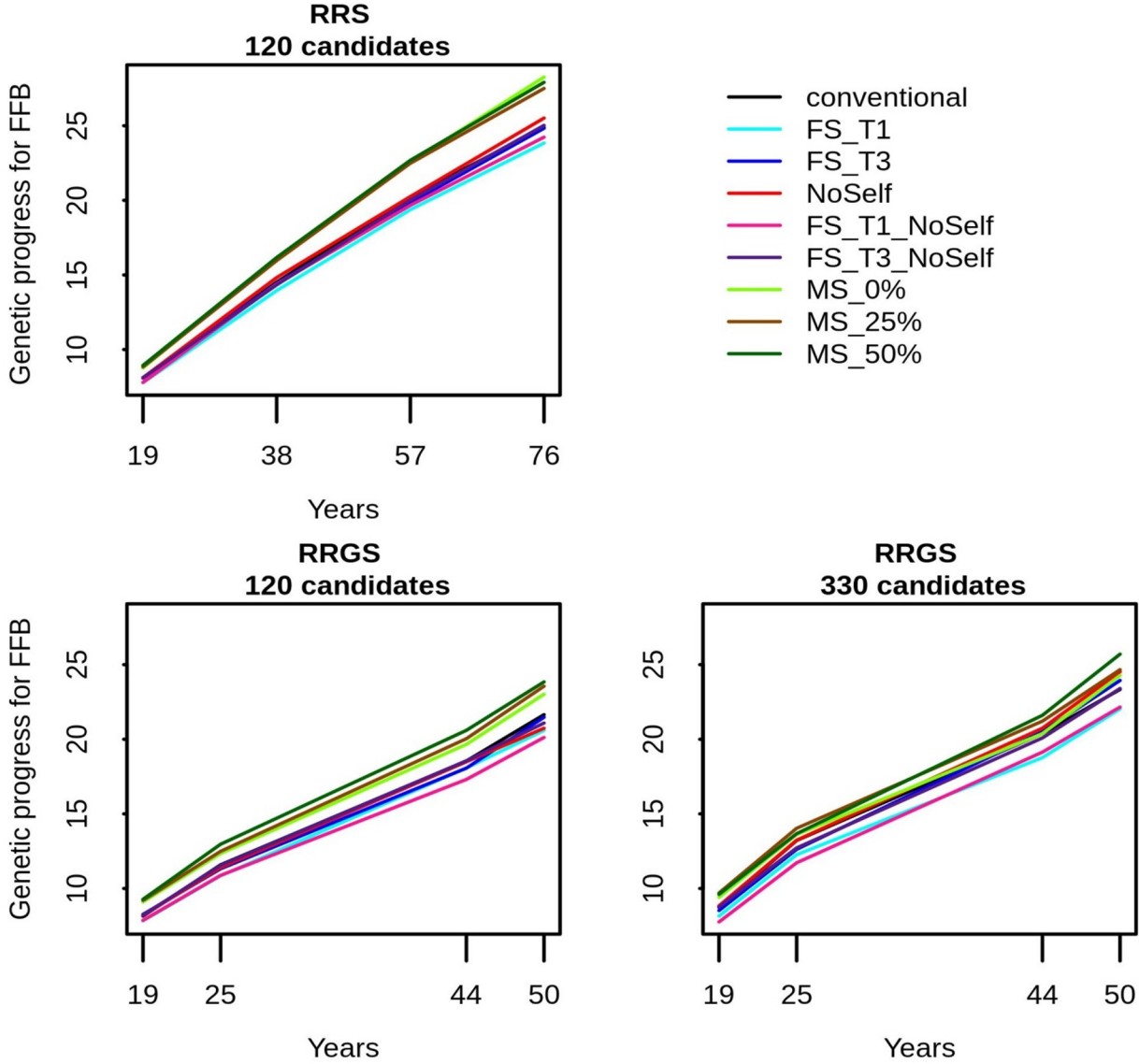

**Fig 4. Genetic progress for FFB (in percentage of hybrids performance at generation 0) in the La Mé population according to breeding scheme (RRS with 120 candidates, RRGS with 120 candidates and RRGS with 330 candidates), years (19 to 76 in RRS and 19 to 50 in RRGS) and methods of selection and mating.** Figures are means over 30 replicates.

full-sib families due to the fixed number of crosses, it increased the maximum possible rank of the selected individuals compared with 120 candidates in RRGS and RRS with all methods except conventional and NoSelf. Checking the minimum rank of the selected individuals showed that MS generally selected the very first individuals, i.e. with rank 1 (not showed), as done, by construction, by the other methods of selection and mating.

For all methods but MS, the number of crosses per selected La Mé individual in all schemes was fixed (see Figs 8 and 9), as specified in the simulation procedure. However, with MS, the number of crosses per selected individual varied widely among individuals, from around 1 (Fig 8) to more than 10 (Fig 9). The distribution of the number of crosses per selected individual was highly skewed towards small values, with the three best individuals that could be involved in more than one half of the crosses (see e.g. S7 Fig).

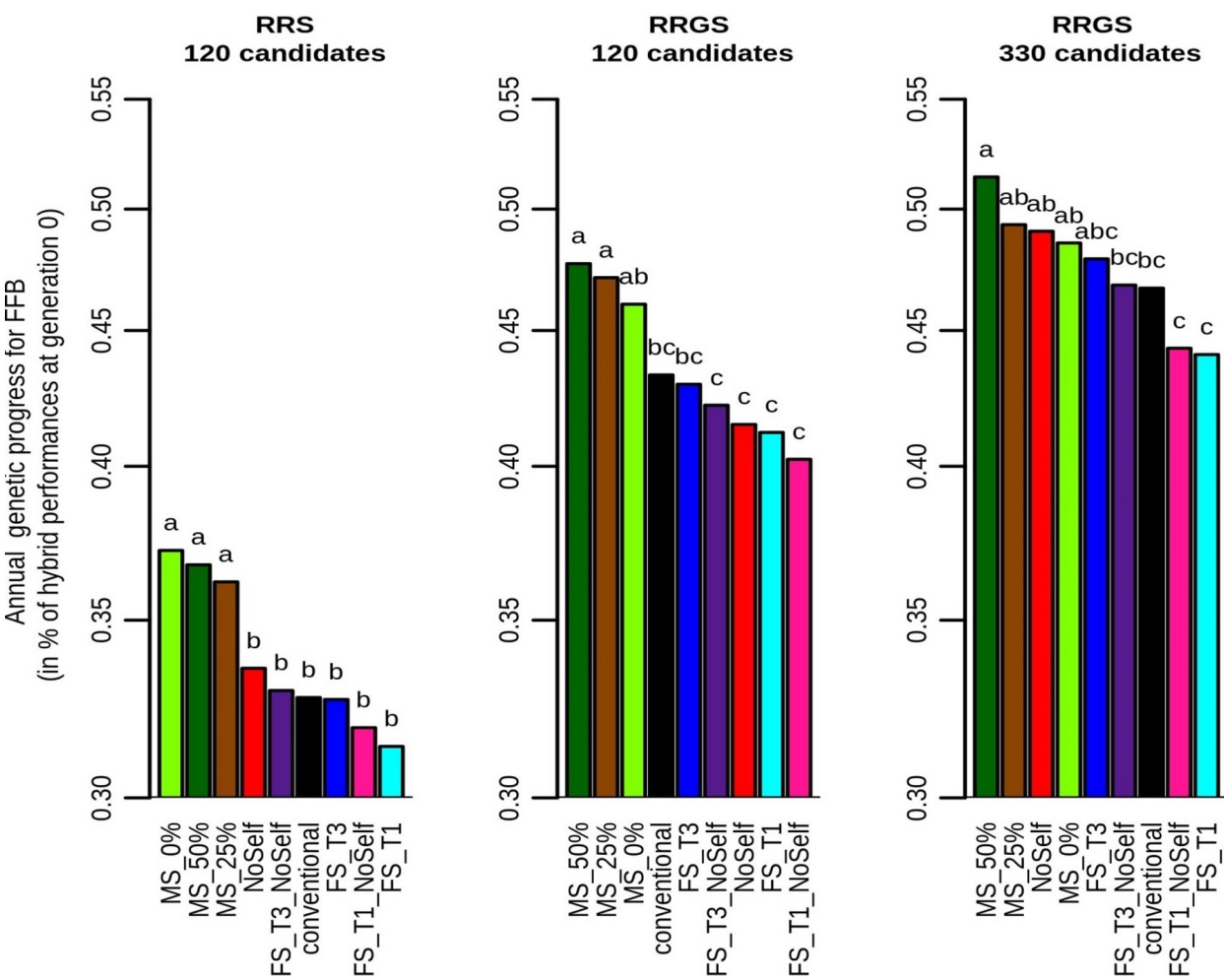

**Fig 5. Annual genetic progress for FFB (in percentage of hybrids performance at generation 0) according to method of selection and mating and breeding scheme (RRS with 120 candidates, RRGS with 120 candidates and RRGS with 330 candidates).** Figures are means over 30 replicates. Values with the same letters are not significantly different within a breeding method at P = 5%.

Mate selection led to a decrease in the maximum number of individuals selected per full-sib family compared to the conventional method, in particular with 330 selection candidates (Fig 10). This is related to the fact, already mentioned above, that MS could select individuals of rank above 16 if needed to fulfil the constraints. However, MS still selected several individuals per full-sib family (by contrast with FS_T1 and FS_T1_NoSelf). The threshold in the inbreeding increase $c_{\Delta F}$ hardly affected the result.

The correlation between the rank of the selected La Mé individuals and the number of times they were crossed to produce the next generation (Fig 6B) was below -0.6 with MS, showing that, with this method, the better the individuals, the more often they were crossed (although the relationship between these two variables was not linear, as showed on S7 Fig). For the other methods, this correlation was close to zero, as expected.

Another mean of investigating the relationship between the genetic value of the selected individuals and the way they were mated is to consider the correlation between the ranks of mates, as illustrated in S8 Fig. For conventional selection, it shows a random pattern, while for mate selection, the pattern is mostly triangular with absence of mates between individuals with

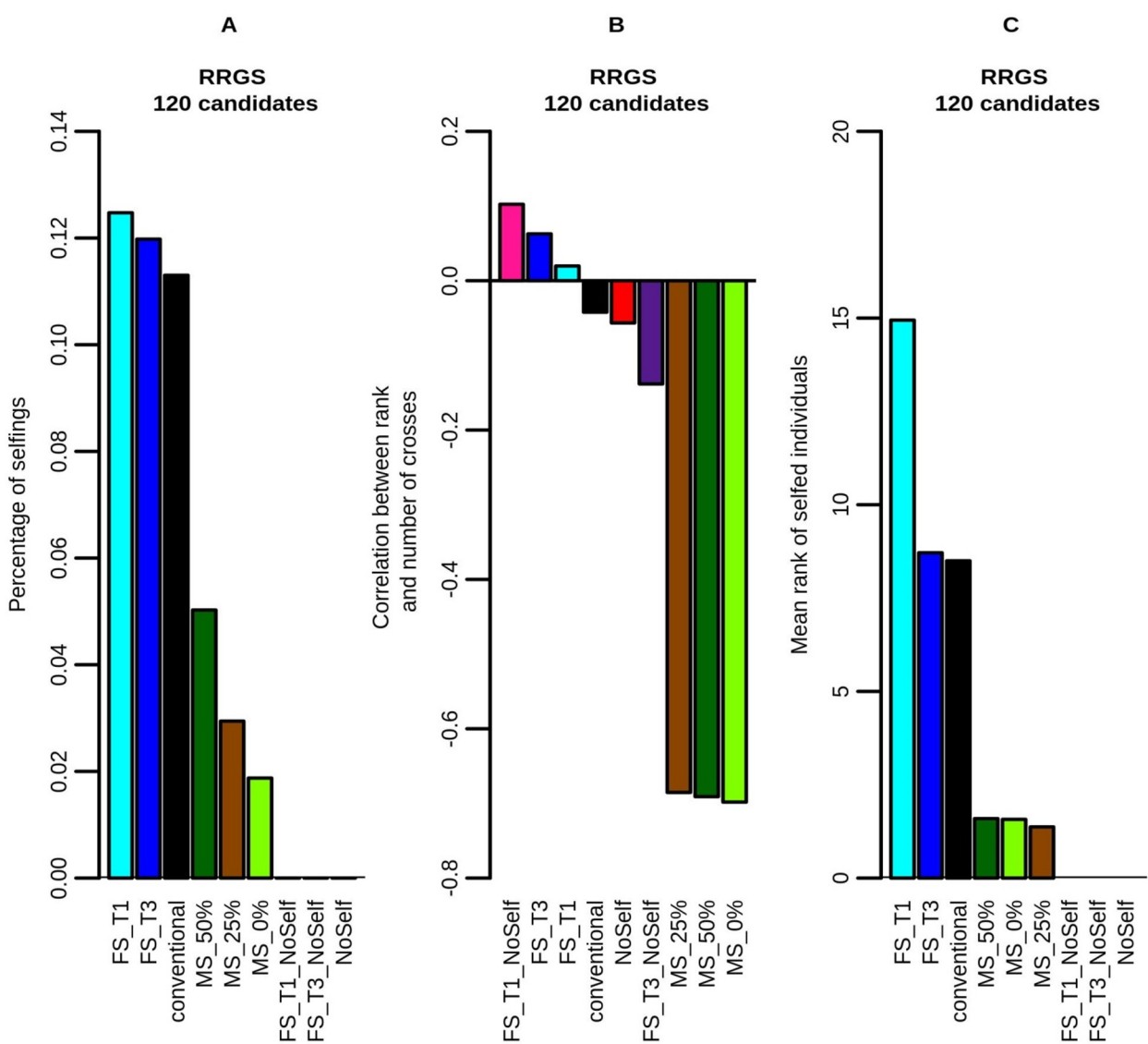

**Fig 6.** Example results obtained with RRGS and 120 selection candidates showing (A) percentage of self-fertilization in the La Mé population, (B) correlation between the rank of the selected La Mé individuals and the number of times they are crossed to produce the next generation and (C) mean rank of selfed individuals. Figures are means over 30 replicates.

low ranks and abundance of mates between individuals with either contrasting ranks or top ranks.

With mate selection, the mean rank of the self-fertilized selected La Mé individuals (Fig 6C) was close to one, i.e. much lower than with the conventional method (with mean rank of the selfed individuals around 8, as expected from random mating among the 16 best individuals). This shows that mate selection preferentially selfed the best individuals. The highest values in mean rank of the selfed individuals was reached when a threshold was set in terms of number of individuals selected per full-sib family, under the effects of random mating and of the fact that individuals of lower performance (i.e. higher rank) had to be selected to reach 16 selected individuals. The mean rank of the selfed individuals increased with the stringency of the threshold (i.e. higher mean rank of selfed individuals with FS_T1 than with FS_T3) and the

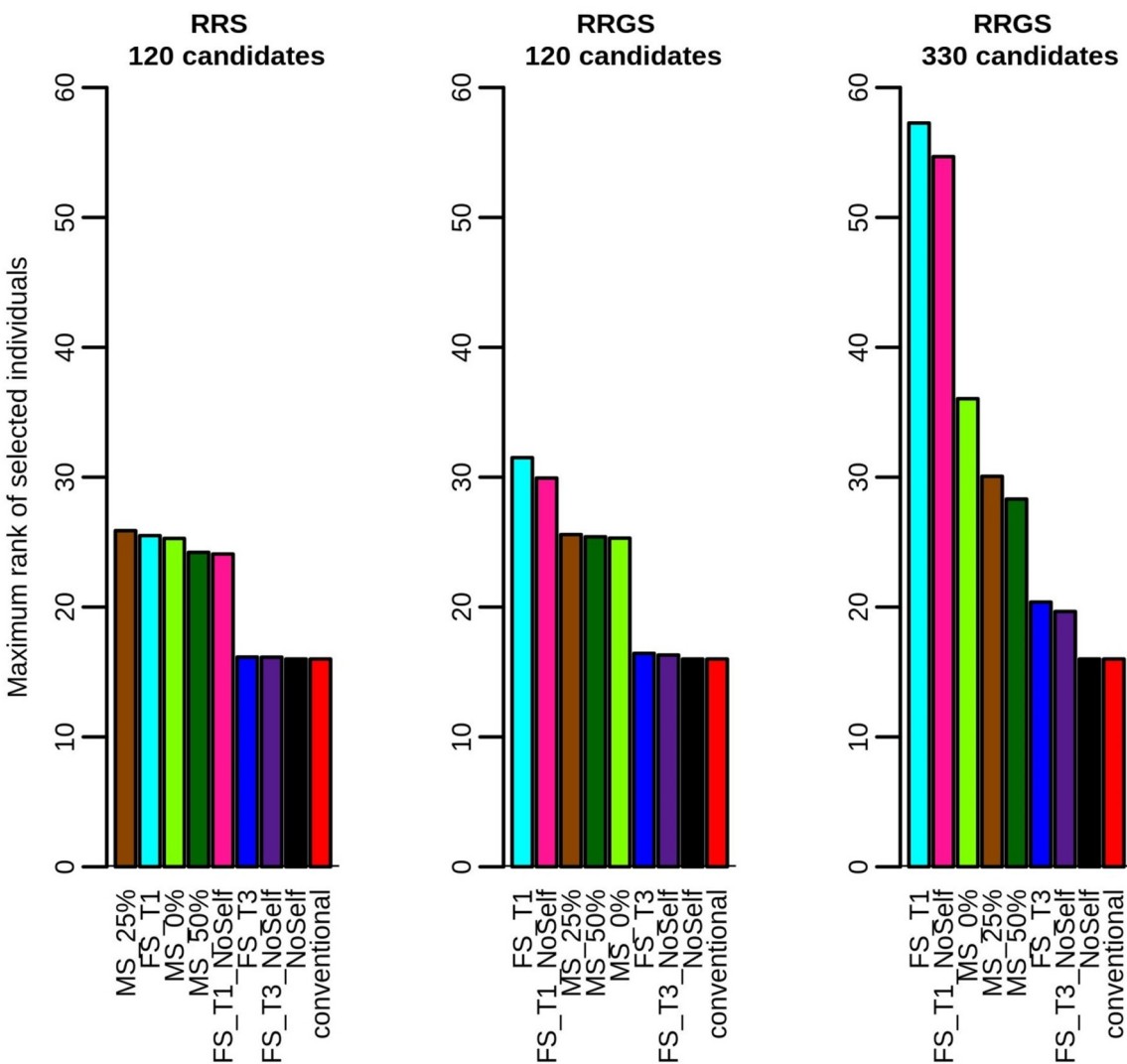

**Fig 7. Maximum rank of selected individuals according to method of selection and mating and breeding scheme (RRS with 120 candidates, RRGS with 120 candidates and RRGS with 330 candidates).** Figures are means over 30 replicates.

number of candidates (i.e. higher mean rank of selfed individuals with 330 candidates than with 120, as the larger size of the full-sib families with 330 candidates led to selecting individuals with even higher rank) (not showed).

The mean relationship between the La Mé individuals crossed with each other to produce the next generation was investigated but no clear pattern was observed in this parameter (not showed).

## Discussion

A forward-in-time simulation with massively parallel processing on a high-performance computing cluster made possible the evaluation of the mate selection approach over the equivalent to several decades and with various scenarios of oil palm breeding. This showed that mate selection will allow oil palm breeders to reduce the rate of increase in inbreeding in the parental populations while maximizing the performances of the hybrids compared to the

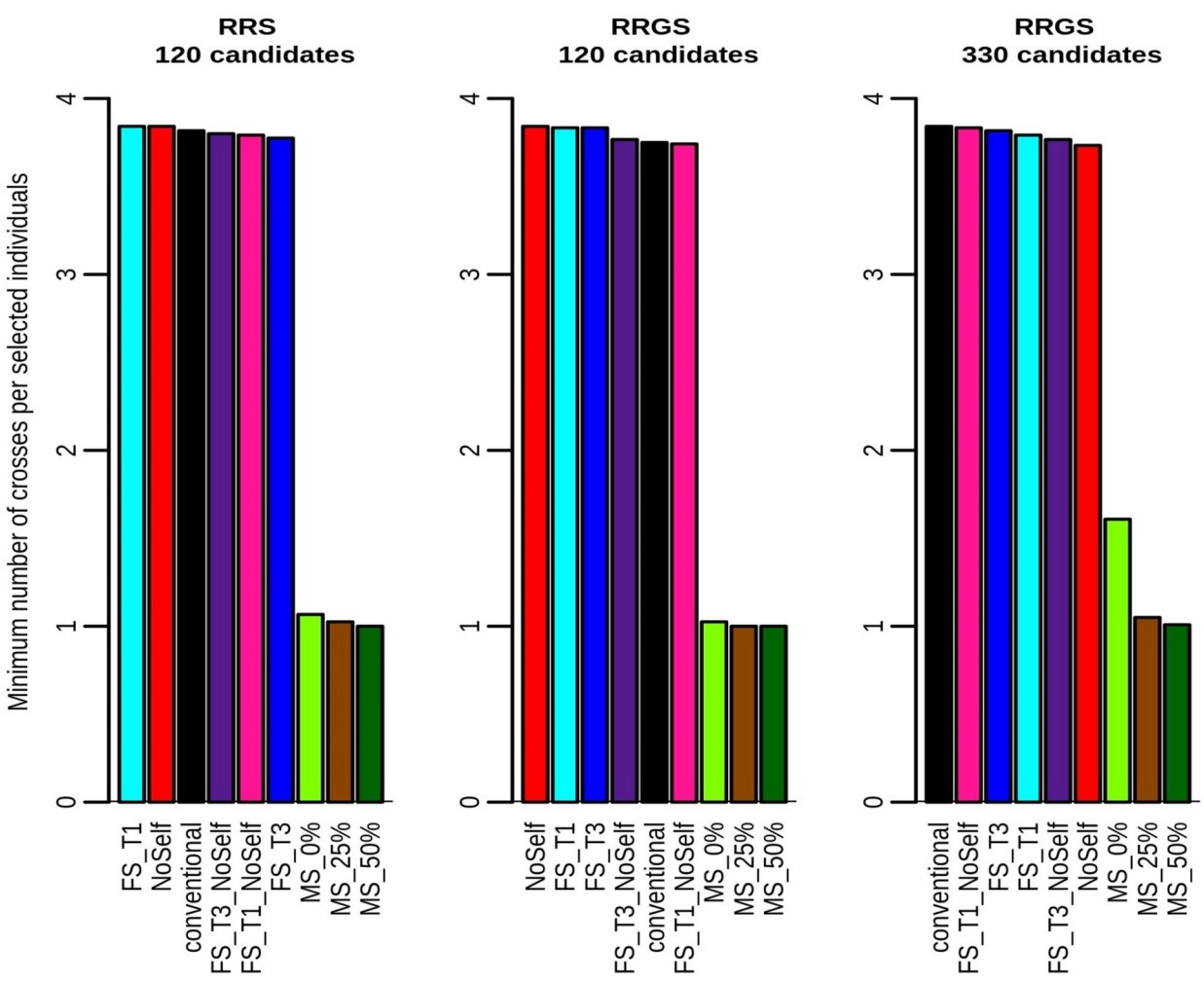

**Fig 8. Minimum number of crosses per selected individuals according to method of selection and mating and breeding scheme (RRS with 120 candidates, RRGS with 120 candidates and RRGS with 330 candidates).** Figures are means over 30 replicates.

conventional method. The change in inbreeding with MS compared to the conventional method ranged from +10.3% to -29.7% when measured with pedigree and from +0.6% to -3.6% when measured by SNPs, while the annual genetic progress was from 3.9% to 13.6% higher with MS, depending on breeding scenarios and MS parameters. Also, for mate selection, we characterized the optimized solution (i.e. the set of selected individuals and their mating design) retained by the simulated annealing algorithm in terms of breeding rules. The optimal solution retained by MS differed by five breeding characteristics from the conventional solution: selected individuals covering a broader range of genetic values, fewer individuals selected per full-sib family, decreased percentage of selfings, selfings preferentially made on the best individuals and unbalanced number of crosses among selected individuals, with the better an individual, the higher the number of times he is mated. By changing the threshold for inbreeding, more stringent solutions could be obtained to favor reduction of inbreeding over increasing of gain. The reduction in inbreeding will lead to a reduced risk of loss of favorable alleles and, for traits with dominance effects, a reduced risk of inbreeding depression. Stronger slowing-down in inbreeding was attained with simple methods of inbreeding management but they were penalised by substantial decreases in genetic progress.

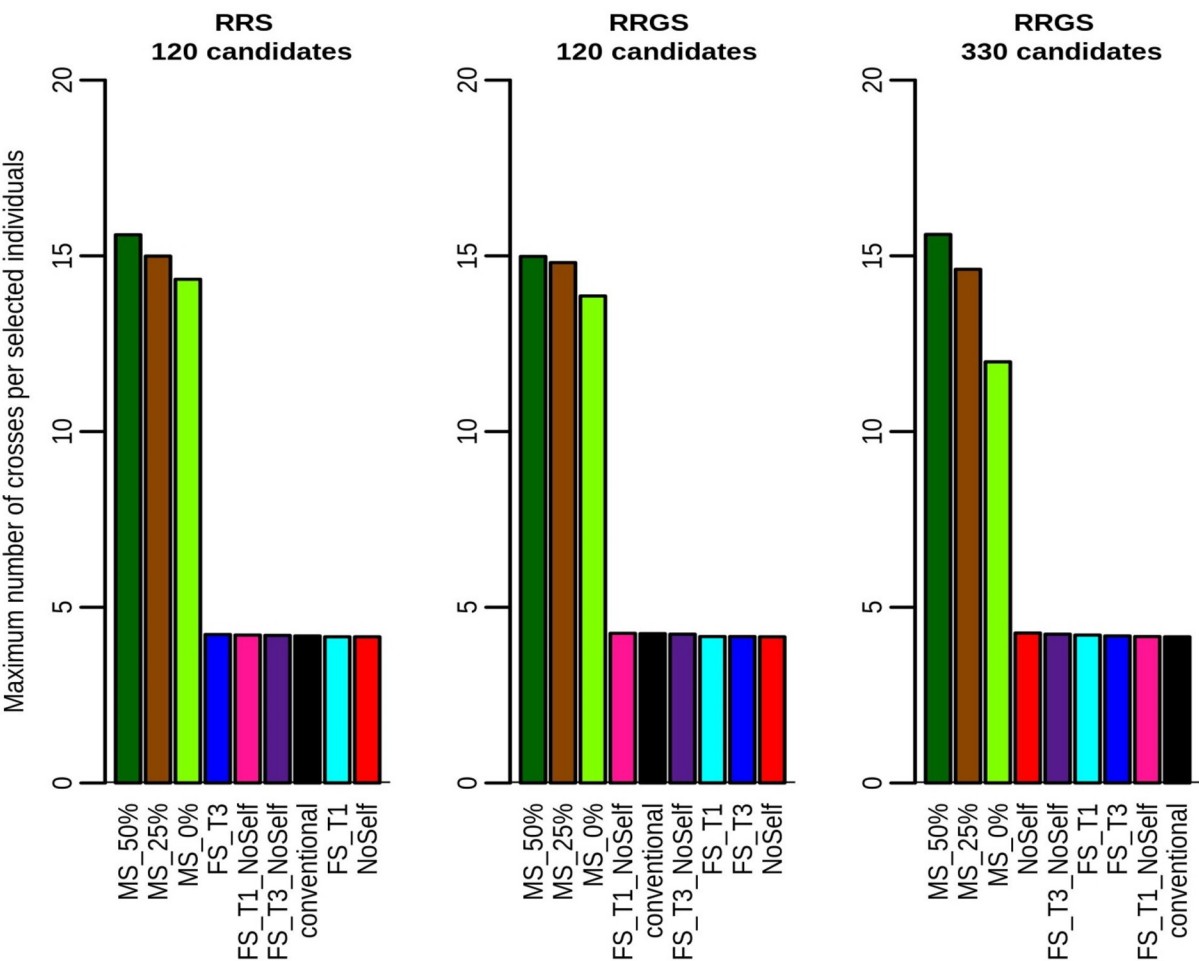

**Fig 9. Maximum number of crosses per selected individuals according to method of selection and mating and breeding scheme (RRS with 120 candidates, RRGS with 120 candidates and RRGS with 330 candidates).** Figures are means over 30 replicates.

A few articles recently focused on the inbreeding management in breeding programs. Like mate selection, AlphaMate [13] optimizes selection and crosses among selected individuals with mating constraints, using a metaheuristic optimization approach. However, it uses an evolutionary algorithm instead of simulated annealing. Simulated annealing is a single solution metaheuristic, i.e. modifying and improving a single candidate solution, while an evolutionary algorithm is a population-based approach, working on multiple candidate solutions. More importantly, AlphaMate differed from the mate selection implemented here in the way it handles inbreeding. Indeed, AlphaMate optimizes genetic gain and/or inbreeding, with inbreeding being included in the evaluation function with a given weight that balances the importance of gain over inbreeding. In our mate selection implementation, inbreeding intervened as a constraint to keep or discard solutions. AlphaMate can therefore also be used for conservation purposes, using an evaluation function that only targets inbreeding, while this is not possible with our approach, which can only optimize genetic gain. However, AlphaMate could not be used for our case study as we focused on the performances of hybrid crosses for a multiplicative trait, which required considering jointly the two parental populations. Lin et al [7] also developed approaches to limit inbreeding and maximize genetic gain. Their approaches optimized mating choosing the crosses with the lowest value of a mate allocation metric

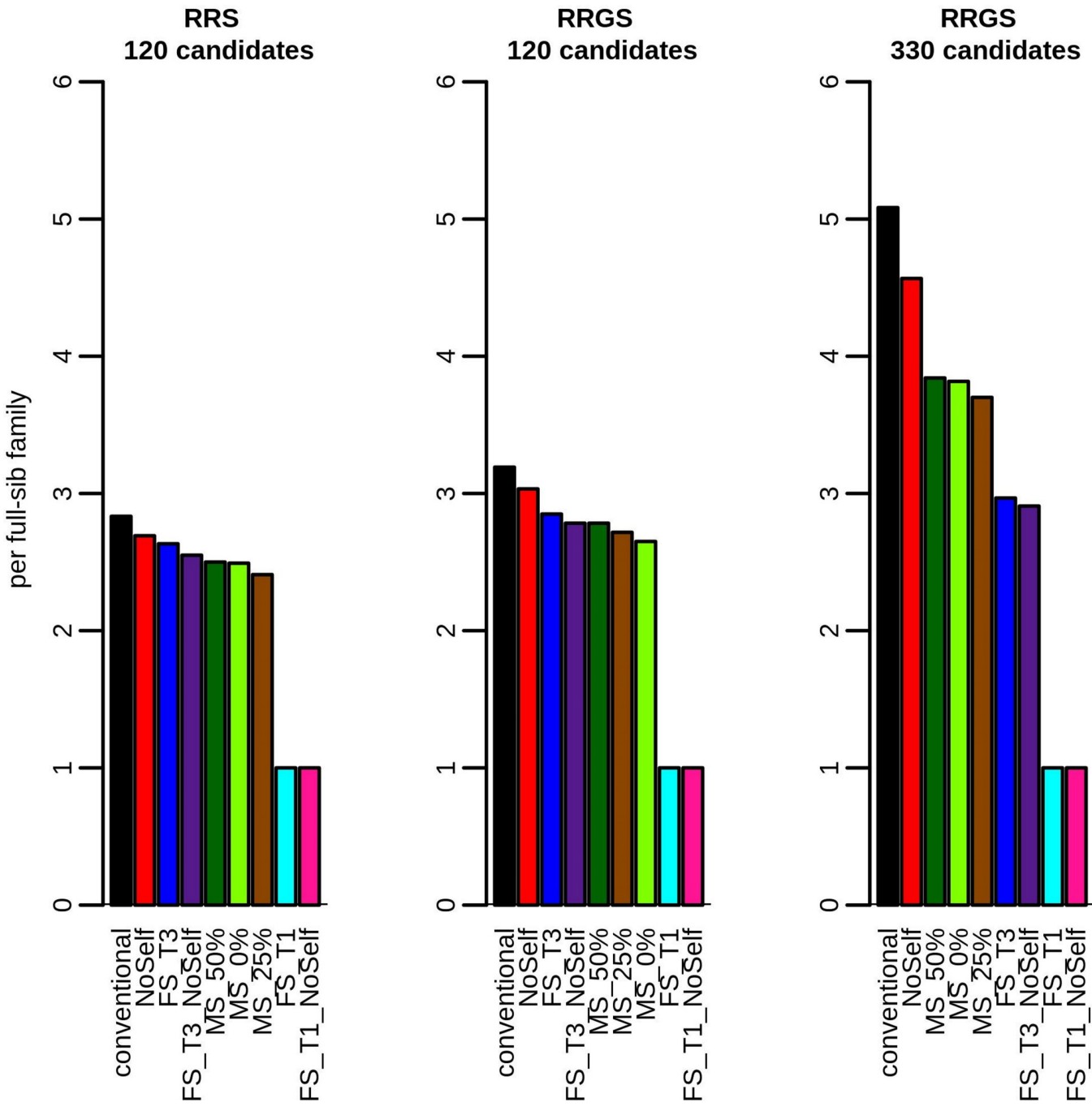

**Fig 10. Maximum number of individuals selected per full-sib family according to method of selection and mating and breeding scheme (RRS with 120 candidates, RRGS with 120 candidates and RRGS with 330 candidates).** Figures are means over 30 replicates.

(computed from the genomic relationships among the selection candidates) and/or optimized selection adjusting the estimated genetic values of the selection candidates by their relationships and using a genetic algorithm to optimize the set of selected individuals. A comparison of these different approaches is lacking. Optimum contribution selection (OCS) [25,39] is another approach to maximize the genetic gain while restricting the rate of inbreeding in the

progeny by constraining the relationships between the selected parents. It optimizes the genetic contribution (i.e. number of matings) of each selection candidate but does not optimize matings among the selected individuals. In the current study, we preferred considering an approach optimizing both selection and matings. A recent study [40] proposes a weighting method to modify the additive relationship matrix used on OCS and promote in turn individuals carrying multiple heterozygous loci, resulting in benefits in short-term genetic gains and higher selection plateau. Although it is a promising alternative to OCS without the computational burden of mate selection, it is not intended to control explicitly inbreeding at a per generation basis. To date, most work on mate selection has focused on animal species, where it has been successful. For example, in aquaculture breeding, mate selection has been shown to be effective in reducing inbreeding while optimizing genetic gain ([41], [42], [43]) and, in the short term, to outperform OCS in controlling inbreeding [41,42]. Similarly, mate selection proved effective in the short term for achieving genetic gains while controlling inbreeding in small flocks [44]. Although these results are similar to those we observed here, the present study differs from these previously published articles in the breeding field. Indeed, they considered single populations, i.e. with the same population used to measure inbreeding and genetic gain, whereas here, inbreeding was of concern in the parental populations and genetic gain was measured in the hybrid population.

We identified five characteristics distinguishing the optimal solution found by the simulated annealing algorithm compared to the solution of the conventional method without optimization. First, with mate selection the selected individuals cover a broader range of genetic values, as mate selection selects the individual ranked first, like conventional selection, but also selects individuals of lower genetic value. Second, there is a reduction in the number of individuals selected per full-sib family. This is related to the first characteristic, as limiting the number of individuals selected per full-sib family necessarily leads to selecting individuals in families of lower mean genetic value. Third, there is a decrease in the percentage of selfings. With mate selection, the percentage of selfing was strongly affected by the coefficient $c_{\Delta F}$ used to control the inbreeding threshold, with the lower the inbreeding threshold, the lower the percentage of selfings. Thus, mate selection retains a higher percentage of selfings when the inbreeding constraint is low (MS_50%), i.e. when more emphasis is given to the optimization of the genetic gain, than when the inbreeding constraint is high (MS_0%). However, the fact that MS_50% led to inbreeding increase close to the conventional method (Fig 3) while its percentage of selfing was lower than with the conventional method indicated that reducing the percentage of selfing is also used by mate selection to optimize the genetic gain, and not only to reduce inbreeding. Fourth, the selfings are preferentially made on the best individuals among the selected ones. Fifth, the number of crosses is highly unbalanced among selected individuals, with a higher number of crosses on the individuals with the best ranks. This point was also noted in animals, where mate selection resulted in a high variance in the number of offspring per breeding animal [44]. The correlation between rank and number of crosses was not affected by $c_{\Delta F}$, suggesting that mate selection adjusted the number of crosses per individual to optimize the genetic gain rather than to limit inbreeding. We noted that the high-ranking parents mate a diversifying panel of individuals in terms of ranking, while the parents with the lowest ranks never mate together. This resembles to a compensatory mating scheme where carriers of favorable alleles preferentially mate non-carriers [40,45]. These simple rules could help breeders to optimize selection and matings without complex analyses. However, the results showed that, for a given breeding scenario (i.e. RRS or RRGS with 120 or 330 candidates), slowing down the inbreeding progression and maximizing the genetic gain were antagonistic, with the highest genetic gain achieved with the lowest constraint on inbreeding. This was in particular the case over the long term, as after several breeding cycles inbreeding with

          

MS could become higher than inbreeding with the conventional method (Fig 1). We hypothesize that this resulted from the fact that with mate selection, the individuals with the highest rankings were crossed with a large number of partners, which generated high proportions of half-sibs in the progeny, thus negatively impacting inbreeding in the following generations despite the constraints applied. Mate selection, as well as classic formulations of OCS, is based on single generation constrains, neglecting long-term consequences. We recommend that breeders use mate selection, with implementation starting with preliminary analyses to identify the proper parameters to reach the goals of the breeding program in terms of inbreeding and genetic gain.

Genealogical inbreeding gives the expected inbreeding over an infinite number of individuals with the same pedigree (full-sibs), while inbreeding computed over SNPs or QTLs corresponds to the actual value for the considered loci and individuals [46]. These two ways of measuring inbreeding also differed as identity-by-descent and identity-by-state are confounded at the marker level [47], leading to potential upward bias. However, the markers used in the simulations comprised unique alleles at the founder phase. Markers, however, are affected by linkage disequilibrium, which in turn impacts drift and inbreeding, and it is that phenomenon that explains the higher values of SNP-based inbreeding compared to genealogical inbreeding. One way to get inferences of identity-by-descent by using markers is through runs of homozygosity (ROH) [47,48], which are contiguous regions of the genome where an individual is homozygous due to parents transmitting identical haplotypes. We investigated this possibility in preliminary analyses using the detectRuns R package [49]. ROH-based genomic inbreeding values were overall intermediate between genealogical inbreeding and SNP-based inbreeding. The results were affected by the parameters and the method used to identify the ROH, as already noted in the literature [48]. Also, the ROH approach largely increased computation time and gave inbreeding values highly correlated with the genomic inbreeding values obtained here (>0.9). As a consequence, we did not use the ROH approach in the present study, although it is commonly used in studies that investigate the history of populations. Regarding inbreeding measured at QTLs, higher values were obtained compared to inbreeding at SNPs. This likely resulted from selection, which amplified the fixation of alleles at QTLs compared to genome regions that did not affect the traits under selection.

The mate selection approach implemented could be improved in several ways, which were not investigated for computational reasons. Here, the optimization was done for the La Mé parental population, while the conventional method was used for Deli. In practice, mate selection should consider the two parental populations jointly. Based on the results we obtained, this would certainly further increase the genetic performance of the hybrids, while also limiting the increase in inbreeding in Deli. This can be achieved by simple changes in the algorithm, using one $X$ matrix for each parental population and applying the replacement function in each population. We initially implemented such an algorithm but this largely increased the convergence time, as the space of possible solutions became much larger. This would have made it impossible to carry out this study, for which, by applying mate selection only to the La Mé, several months of calculations were necessary to obtain the results for all the repetitions and scenarios. Therefore, we finally chose the approach presented here, only considering the La Mé population. Other changes could be to allow the method to adjust the number of individuals selected and the total number of crosses between selected individuals, instead of considering these two parameters as fixed [50]. Another possible improvement of the method would be to allow variations in the number of individuals per cross between selected individuals (i.e. the number of full-sibs), instead of considering it fixed. In this case, the $X$ matrix would no longer contain 0s and 1s but integer values, and the mating constraint on the number of crosses would be replaced by a constraint on the total number of progeny individuals.

The replacement function would then replace individuals between crosses instead of replacing crosses. This would give an additional level of flexibility to the method but would also largely increase the space of possible solutions, and thus convergence time.

The present work has shown that, in terms of annual genetic progress, breeding schemes that integrate GS outperform conventional breeding schemes, which was in agreement with previous articles [17,33]. Thus, RRGS led to largely higher levels of annual genetic progress than RRS and, as expected, the superiority of RRGS increased with the number of selection candidates. The higher level of annual genetic progress with RRGS was observed despite a lower genetic progress after 4 cycles compared to RRS. This is due to the fact that, in oil palm, selection based on progeny tests is more accurate than GS, as previously reported [17,33,51]. However, the decrease in the number of years required for four cycles is stronger than the reduction in prediction accuracy, leading to a greater annual genetic progress with RRGS than with RRS. The higher level of inbreeding at SNPs after four cycles in RRS compared to RRGS with 120 candidates might be a consequence of the higher selection accuracy of RRS and the characteristics of oil palm breeding populations. These populations are highly structured (i.e. there are small number of full-sib families) and have limited genetic variation within families due to low heterozygosity in parents. In this context, high levels of selection accuracy ($r \sim 0.8$–$0.9$ with oil palm progeny-tests [17]) will clearly discriminate between families and, without a strategy of inbreeding control, will tend to select several individuals from a limited number of the best families. In contrast, the lower selection accuracy of GS ($r \sim 0.6$–$0.7$ [33]) will tend to select individuals from a larger number of full-sib families, due to a lower ability to capture differences between families.

Here, we considered a multiplicative trait with heterosis explained by a model without dominance, by the product of purely additive and complementary components between parents [23,24]. However, even low levels of heterosis in the basic components of the multiplicative trait can lead to high levels of heterosis in the latter [23]. This possibility was not addressed in the present article. It would be interested to consider, in a future simulation study, BN and BW traits with varying levels of dominance, in order to evaluate whether this affects the performance of the different methods of selection and mating compared here. Another possible extension of the present study would be to model, in addition to BW and BN, the reproductive traits that could be affected by inbreeding depression in the parental populations. Indeed, as mentioned above, inbreeding depression in parental populations is not a problem in terms of hybrid performance (as hybrid crossings restore a high level of heterozygosity), but it might be detrimental for reproductive traits that affect seed production, like seed germination. It would therefore be interesting to include in the simulation these reproductive traits and to model them including dominance effects, in order to investigate in greater detail how they could be affected in the parental populations by the different strategies tested here.

In conclusion, this study showed that, compared to the conventional method without optimization, mate selection can significantly decrease inbreeding in parental populations and increase annual genetic progress in hybrids, with the magnitude of the effect depending on mate selection parameters and breeding scenarios. We also identified five characteristics distinguishing the optimal solution found by the simulated annealing algorithm compared to the solution of the conventional method without optimization. We recommend that breeders use mate selection, with preliminary analyses to identify the proper parameters to reach the goals of the breeding program in terms of inbreeding and genetic gain. This study also confirmed the superiority of RRGS over RRS in oil palm. Prospects include implementing various improvements to the method presented here, comparing it with recent software that also aims to limit inbreeding and maximize genetic gain, and simulating dominance effects.

## Supporting information

**S1 Fig. Example result of evolution of evaluation values and temperature according to iterations, until convergence.**
(DOCX)

**S2 Fig. Inbreeding computed from SNPs in the La Mé population according to the generations (0–4), breeding methods (RRS with 120 candidates, RRGS with 120 candidates and RRGS with 330 candidates) and methods of selection and mating.** Boxplots show distribution of values over 30 replicates.
(DOCX)

**S3 Fig. Genealogical inbreeding in the La Mé population according to the generations (0–4), breeding methods (RRS with 120 candidates, RRGS with 120 candidates and RRGS with 330 candidates) and methods of selection and mating.** Figures are means over 30 replicates.
(DOCX)

**S4 Fig. Genealogical inbreeding in the La Mé population after four breeding cycles according to method of selection and mating and breeding scheme (RRS with 120 candidates, RRGS with 120 candidates and RRGS with 330 candidates).** Figures are means over 30 replicates. Values with the same letters are not significantly different within a breeding method at P = 5%.
(DOCX)

**S5 Fig. Inbreeding at QTLs for trait 1 in the La Mé population according to the generations (0–4), breeding methods (RRS with 120 candidates, RRGS with 120 candidates and RRGS with 330 candidates) and methods of selection and mating.** Figures are means over 30 replicates.
(DOCX)

**S6 Fig. Genetic progress for FFB (in percentage of hybrids performance at generation 0) in the La Mé population according to breeding methods (RRS with 120 candidates, RRGS with 120 candidates and RRGS with 330 candidates), years (19 to 76 in RRS and 19 to 50 in RRGS) and methods of selection and mating.** Boxplots show distribution of values over 30 replicates.
(DOCX)

**S7 Fig. Example result of number of crosses per selected La Mé individual according to their rank, with mate selection.**
(DOCX)

**S8 Fig. Example result showing the rank of the parent used as male and the rank of the parent used as female for the 32 crosses among the selected La Méindividuals, with mate selection and conventional selection.**
(DOCX)

## Acknowledgments

We acknowledge the CIRAD-UMR AGAP HPC data center of the South Green bioinformatics platform (http://www.southgreen.fr/) for their help. We acknowledge the CETIC (African Center of Excellence in Information and Communication Technologies) for its support. We

thank Benoit Cochard and Florence Jacob (PalmElit) for discussions, and Marie Pégard (INRAE) for advice regarding parallel computing and discussions.

## Author Contributions

**Conceptualization:** Leopoldo Sanchez, David Cros.

**Formal analysis:** Billy Tchounke, David Cros.

**Funding acquisition:** Leopoldo Sanchez, David Cros.

**Investigation:** Billy Tchounke, David Cros.

**Methodology:** Leopoldo Sanchez, David Cros.

**Project administration:** Joseph Martin Bell, David Cros.

**Software:** Billy Tchounke, David Cros.

**Supervision:** Joseph Martin Bell, David Cros.

**Validation:** Billy Tchounke, David Cros.

**Visualization:** Billy Tchounke, David Cros.

**Writing – original draft:** Billy Tchounke, David Cros.

**Writing – review & editing:** Leopoldo Sanchez, Joseph Martin Bell, David Cros.

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
