## [Decision Letter · Decision Letter 0]

4 Apr 2023

Dear Dr. Cros,

Thank you very much for submitting your manuscript "Mate selection: a useful approach to maximize genetic gain and control inbreeding in genomic and conventional oil palm (Elaeis guineensis Jacq.) hybrid breeding" for consideration at PLOS Computational Biology.

As with all papers reviewed by the journal, your manuscript was reviewed by members of the editorial board and by several independent reviewers. In light of the reviews (below this email), we would like to invite the resubmission of a significantly-revised version that takes into account the reviewers' comments.

We cannot make any decision about publication until we have seen the revised manuscript and your response to the reviewers' comments. Your revised manuscript is also likely to be sent to reviewers for further evaluation.

Sincerely,

Ilya Ioshikhes

Section Editor

PLOS Computational Biology

Reviewer's Responses to Questions

**Comments to the Authors:**

Reviewer #1: Comments to authors

In general, the paper is well written, and most contents are easy to follow. The paper investigated approaches of mate selection using computer simulations. Multiple breeding scenarios were tested, including a scenario with mate selection using simulated annealing optimization algorithms (SA), and some other conventional approaches. The paper found the mate selection using SA resulted in higher genetic gain and less increment of inbreeding across cycles, and provided reasonable discussions.

There is a main issue of the paper that needed to concern. In the paper, genetic gain of the target trait was tested in hybrid population. However, traits in this study were only simulated with additive genetic architecture. So, the paper is missing the power of capturing heterosis in hybrid population. Readers may be more interested in results from a simulation that both additive and dominance effects are simulated for traits (heterosis is from over-dominance). Please at least add a paragraph in the Discussions to give details that recognizing the lack of dominance in the study, and the potential influences on the results, etc.

In addition, the paper did not give info that whether the simulated annealing optimization algorithm was coded in-house or from published package. Please add some info in Methods. and if coded in-house, would be great to make it public for people to re-use in practise.

Overall, the manuscript needs intermediate level of revision, especially on Methods and Results. Details as below.

P5 line 18, marker-assisted selection (MAS) is a term for a specific approach that applied couple decades ago, and in theory, genomic selection is different to MAS. Please remove marker-assisted selection to avoid misleading. The comment applies across the manuscript.

P10 line 124, remove ‘indeed’

P11 line 143, please state what items were compared between scenarios, e.g. genetic gain of target trait and inbreeding level between breeding scenarios.

P12 line 167, ‘R-ASReml’ change to ‘ASReml-R’

P12 line 187-190, please give more detail on how you sampled QTL effects, e.g. effects sampled from normal distribution? QTL positions sampled randomly from SNPs?

And, you only mentioned the traits were simulated following additive genetic architecture (line 157). Please add more detail of how phenotypes simulated, e.g. genetic variance + error term, and the error term evaluated from heritability, etc.

In addition, it is better to show the genetic correlation between the traits after ‘pleiotropic QTLs’ (line 189), rather than putting those info in Suppl. Table.

P13 line 196, for readers’ conveniences and making the paper self-contained, please add a figure to make breeding schemes visualized in the paper, rather than simply referring to other papers.

P15 line 257, ‘matrix X’ changes to ‘matrix X’ (‘X’ in bold). Please revise any matrix notations to bold across the manuscript.

P15 line 263, Let’s change to ‘Let’

P15 line 264, please revise notations, e.g. ‘DiDj’ should be ‘DiDj’ (subscript for ‘i’ and ‘j’). This comment applies to others across the manuscript (e.g. LMiLMj, etc.)

P16 line 276, ‘five first iterations’ changes to ‘first five iterations’.

P18 line 317 - 319, the contents repeated in the table, can be removed from the manuscript.

P19 line 342, ‘calcInbreeding’ is not a default function in R, please give details of package.

P19 line 358-359. Suppl. Figure 3 came before Suppl. Figure 2 in the text of manuscript, please swap.

In addition, the colours used in all figures are hard to be distinguished, especially in line charts, e.g. Fig 1 ‘RRGS 330 candidates’. Please re-plot all figures with colours easy to tell apart.

P20 line 390, the sentence can be moved to line 389 as one paragraph.

P21 line 414-476, the whole section of ‘selection and mating’ in Results is lack of info in Methods, which making it hard to follow. Please add some info in Methods, e.g. what are the items and how they were analysed, etc.

In addition, some contents in this section should belong to Discussions, e.g. P23 line 452.

Please revise the whole section, and relocate some contents to Methods / Discussions.

P25 line 500-519, the paragraph could be moved to the end of Discussions

P25 line 507 -509, the info is vague, better to give details of how long the convergence time if optimizing both the La Me and Deli population against the current one optimizing La Me only.

P28 line 579 – 596, the paragraph could be removed. The core of the study is comparing mate selection with other conventional approaches. However, this paragraph is about different ways of measuring inbreeding, and non-related to your objectives.

P28 line 597 – 599, can be moved to the end of line 596 (or removed).

Please add two paragraphs at the end of Discussions, one regarding the simulation of dominance (as mentioned before), and one to formally conclude the paper (conclusion, ideas for further studies, etc.)

Suppl Table 1. The contents are important for simulations. Strongly recommend to put it in the main manuscript, rather than in supplementary.

Reviewer #2: Mate selection (MS) program have been studied extensively especially in animal breeding programs, so the concept itself is not new. This is nicely written paper addressing the important question of balancing inbreeding and genetic gain in the long-term conventional breeding and molecular breeding schemes of oil palm. There are, however, some limitations of this study as indicated in my comments below. I have several questions and suggestions for the authors to consider:

1. I suggest including a diagram showing the simulated scheme as the description is somewhat confusing for readers who are not familiar with this crop.

2. MS scheme was shown to result in higher inbreeding and higher genetic gain (Figures 2 and 3), for a largely additive genetic trait, over the longer term. It is important to note that with a given inbreeding level, the deleterious load could be different for different traits – hence the genetic gain will vary. Therefore, it is not realistic to generalize the utility of MS scheme for economic traits that are influenced by non-additive gene action. The question is how to design a MS scheme for a breeding program where some of the key traits are controlled by different gene actions (e.g., additive and non-additive).

3. The interpretations of results in this study are mainly focused on comparing different mating and selection schemes. I would suggest authors to also highlight and compare the results between RRS and RRGS schemes.

4. Empirical data from commercial species suggest that inbreeding per generation (and per year) can increase faster with GS schemes, but apparently SNP-based inbreeding in this study was higher for RRS scheme compared with RRGS for a given size of selection population (N=120; Figures 1 and 2). Please clarify/discuss these results.

5. Figures 3 and 4: For the same size of selection candidates (N=120), why the genetic progress after 4 cycles of breeding is higher for RRS scheme compared to RRGS, although the gain per year is obviously higher for the RRGS schemes. Does it mean that RRGS is less efficient/accurate than RRS for the traits investigated in this study?

6. Figures 1 and 2: For the two RRGS schemes, why is inbreeding higher for the larger selection population of 330 compared to 120 candidates? Unless I have missed something, I find these results counterintuitive. Please clarify.

7. Figures 3 and 4: Genetic progress is higher for the increased levels of inbreeding. Perhaps it shows that the trait investigated in this study is not affected by inbreeding depression – hence not a good trait to find a balance between inbreeding and genetic gain, which is perhaps one of the main focuses of this study. A simulated inbreeding management strategy would perhaps allow discrimination between 2 individuals, with same genomic inbreeding coefficient, based on the amount of deleterious load each carry. Please provide your thoughts on the choice of trait in this context.

8. Line 593-595: Genealogical inbreeding levels were very low compared with those computed over SNPs, but perhaps the realized values could be somewhere in-between as the latter are upwardly biased. I guess it might be very useful to present ROH-based genomic inbreeding level even if these are highly correlated with SNP-level inbreeding values – just to check the magnitude of inbreeding.

9. Figures 6 and 7: For the MS scheme, the maximum rank of selected individuals, and the number of crosses per selected individual, was not affected by the inbreeding constraint () when the size of selection population was 120 (as opposed to 330). Why only the larger selection population size is impacting this variable? Does your statement (Line 452-254) “mate selection adjusted the number of crosses per individual to optimize the genetic gain rather than to limit inbreeding” suggest that MS is not the right tool to balance inbreeding and genetic gain in the long-term for this breeding program?

10. Supplementary Table 1: Does the value for additive variance represent interpopulation or intrapopulation variance? Please clarify.

11. I will encourage authors to draw on the learnings from applications of mate selection schemes in animal breeding programs to interpret the results from this study. The patterns of changes in inbreeding level and genetic gain would be somewhat comparable between the plant breeding and animal breeding schemes.

**Have the authors made all data and (if applicable) computational code underlying the findings in their manuscript fully available?**

Reviewer #1: **No: **Already mentioned in comments above

Reviewer #2: **No: **Data availability is subjected to the approval from an agency (PalmElit) which perhaps provides funding for the oil palm breeding program.

PLOS authors have the option to publish the peer review history of their article (what does this mean?). If published, this will include your full peer review and any attached files.

Reviewer #1: **Yes: **Zibei Lin

Reviewer #2: No

Figure Files:

Data Requirements:

Please note that, as a condition of publication, PLOS' data policy requires that you make available all data used to draw the conclusions outlined in your manuscript. Data must be deposited in an appropriate repository, included within the body of the manuscript, or uploaded as supporting information. This includes all numerical values that were used to generate graphs, histograms etc.. For an example in PLOS Biology see here: http://www.plosbiology.org/article/info:doi%2F10.1371%2Fjournal.pbio.1001908#s5.
---

## [Decision Letter · Decision Letter 1]

31 Jul 2023

Dear Dr. Cros,

We are pleased to inform you that your manuscript 'Mate selection: a useful approach to maximize genetic gain and control inbreeding in genomic and conventional oil palm (Elaeis guineensis Jacq.) hybrid breeding' has been provisionally accepted for publication in PLOS Computational Biology.

Best regards,

Ilya Ioshikhes

Section Editor

PLOS Computational Biology

Reviewer's Responses to Questions

**Comments to the Authors:**

Reviewer #1: All my comments have been soundly addressed. Thanks for the revision.

Reviewer #2: Authors have satisfactorily addressed my suggestions/comments.

**Have the authors made all data and (if applicable) computational code underlying the findings in their manuscript fully available?**

Reviewer #1: None

Reviewer #2: **No: **Authors have provided a contact email address if someone would like to access the data and code.

PLOS authors have the option to publish the peer review history of their article (what does this mean?). If published, this will include your full peer review and any attached files.

Reviewer #1: **Yes: **Zibei Lin

Reviewer #2: No

---

## [Editor Report · Acceptance letter]

21 Aug 2023

PCOMPBIOL-D-22-00885R1 

Mate selection: a useful approach to maximize genetic gain and control inbreeding in genomic and conventional oil palm (Elaeis guineensis Jacq.) hybrid breeding

Dear Dr Cros,

I am pleased to inform you that your manuscript has been formally accepted for publication in PLOS Computational Biology. Your manuscript is now with our production department and you will be notified of the publication date in due course.

With kind regards,

Dorothy Lannert
